# Norovirus: An Overview of Virology and Preventative Measures

**DOI:** 10.3390/v14122811

**Published:** 2022-12-16

**Authors:** Natalie Winder, Sara Gohar, Munitta Muthana

**Affiliations:** University of Sheffield, South Yorkshire S10 2TN, UK

**Keywords:** norovirus, epidemiology, outbreak prevention, genome, genotype, classification, control measures, pandemics and transmission

## Abstract

Norovirus (NoV) is an enteric non-enveloped virus which is the leading cause of gastroenteritis across all age groups. It is responsible for around 200,000 deaths annually and outbreaks are common in small communities such as educational and care facilities. 40% of all NoV outbreaks occur in long-term and acute-care facilities, forming the majority of outbreaks. Nosocomial settings set ideal environments for ease of transmission, especially due to the presence of immunocompromised groups. It is estimated to cost global economies around £48 billion a year, making it a global issue. NoV is transmitted via the faecal-oral route and infection with it results in asymptomatic cases or gastrointestinal disease. It has high mutational rates and this allows for new variants to emerge and be more resistant. The classification system available divides NoV into 10 genogroups and 49 genotypes based on whole amino acid sequencing of VP1 capsid protein and partial sequencing of RdRp, respectively. The most predominant genotypes which cause gastroenteritis in humans include GI.1 and GII.4, where GII.4 is responsible for more extreme clinical implications such as hospitalisation. In addition, GII.4 has been responsible for 6 pandemic strains, the last of which is the GII.4 Sydney (2012) variant. In recent years, the successful cultivation of HuNoV was reported in stem cell-derived human intestinal enteroids (HIEs), which promises to assist in giving a deeper understanding of its underlying mechanisms of infection and the development of more personalized control measures. There are no specific control measures against NoV, therefore common practices are used against it such as hand washing. No vaccine is available, but the HIL-214 candidate passed clinical phase 2b and shows promise.

## 1. Introduction

Viruses are microscopic organisms considered ‘transportation shells’ which contain the genetic information solely required for them to survive, thrive, and replicate within a host species causing disease. Many viruses over the year have mutated and adapted to become more efficient at surviving within and outside of a host environment, hence placing immense pressure on our healthcare services. One virus which is exceptional at this is Norovirus (NoV), previously referred to as, the Norwalk virus. It is an endemic virus which has been circulating in the human population for almost 50 years [1,2]. Ever since its discovery, NoV has been recognised as the primary source of gastroenteritis and has contributed to roughly 200,000 deaths annually (predominantly in underdeveloped countries). NoV outbreaks tend to be more common during the winter due to its ability to thrive in cooler temperatures [3]. Once infected a short incubation period of 10–51 h will lead to primary symptoms consisting of stomach cramps, diarrhoea, and vomiting with a tendency to originate and spread within small communities, which usually last around 2–3 days [4,5]. The pathogenesis of Nov suggests a broadening and blunting of the intestinal villi, crypt-cell hyperplasia and cytoplasmic vacuolization of the small bowel leading to the previously described symptoms. This gives rise to serious risk of death within the elderly population (65+) with an estimated 80 deaths occurring annually. More importantly, NoV outbreaks in the UK in recent years have shown that NoV is a causative agent of both a clinical and economic burden [6,7,8].

However, over the last few years with the global pandemic (COVID-19), NoV reported cases in the UK decreased significantly as have other infections, likely due to the overburdened healthcare services, which may have led to unchecked surveillance and testing. However, since the COVID-19 restrictions and measures were relaxed there has been a sudden increase in reported cases of NoV [8]. As COVID-19 transmission rates reduce following successful vaccination programs, NoV cases have begun to rise once more with a 48% higher incidence of reported cases than expected by the UK Health Security Agency (UKHSA) [9]. Fortunately, reported cases have stayed below pre-pandemic threshold but are indicative of yet another disease which would impose a high financial and social burden. Therefore, controlling the transmission of NoV outbreaks as well as preventing its resurgence is crucial, but in practice exceedingly difficult. Due to NoV having a rapid mutation rate, interventions such as vaccines and antivirals have proven difficult to manufacture.

Currently, the most efficient preventative method against NoV transmission involves actively performing hand hygiene practices as per government advise [10]. In this literature review, we will explore NoV virology and its outbreaks, while presenting evidence on the effects of current governmental guidelines for protecting the public from a new pandemic, and exploring new potential preventative NoV outbreak measures.

## 2. Norovirus (NoV) Virology

### 2.1. Genome and Structure

NoV is an enteric non-enveloped virus belonging to the Caliciviridae family and Norovirus genus [1,4]. It is estimated to be 27–35 nm in diameter, and its virion is highly stable under extremely hostile conditions, including a range of pHs (pH 3 to 7) and temperature as high as 60 °C, indicating the difficulty of eradicating and controlling its spread using routine methods [11]. Its genome was first characterised in 1990 using a Norwalk virus, and was later sequenced and labelled in 1993 [12]. NoVs are considered to be genetically diverse and are classified into multiple genogroups, genotypes, polymerase groups (P-groups) and P-types. The classification of NoV into these different subtypes is achieved via complete amino acid and partial nucleotide sequencing of VP1 and RdRp, respectively [13]. This system allows us to continuously develop our knowledge of NoV and its newly emerging strains and will be further discussed [13].

The majority of NoVs harbour a genome organised into 3 overlapping open reading frames (ORFs), with the exception of murine NV (MNV-1) which contains a fourth ORF [12,14]. ORF1 is 5100 bases long and encodes a structural polyprotein, which is processed into 6 non-structural proteins upon post-translational cleavage by viral protease, NS6 (Figure 1) (Table 1) [12]. Among those 6 non-structural proteins is NS7, a 3D-like RNA-dependent RNA polymerase(RdRp), which is responsible for viral RNA replication. ORF2 is 1600 bases long and encodes for viral protein 1 (VP1,) a major capsid protein (Figure 1) (Table 1) [12]. VP1 is organised into the shell (S) domain and protruding 1 (P1) and P2 subdomains of the P domain (Figure 2) [12,15]. In capsid form, the S domain surrounds the viral RNA and is linked to the P domain (P1 and P2) via a flexible hinge [16].

Furthermore, the P2 subdomain is a hypervariable region and the most-surface exposed part of the viral particle, which harbours the histo-blood group antigens (HBGA) binding interface and “major neutralising epitopes for immune evasion and cell infiltration” [17]. In addition, this region is characterised by a high mutational frequency, which is likely a result of host-immunity selective pressure and thought to contribute to the rapid emergence of new strains [18,19]. Each viral particle is composed of 90 dimers of VP1 assembled in a T = 3 icosahedral symmetry, with P2 subdomain being the most exposed part [1,15,18,20].

ORF3 (Figure 1) (Table 1) is 720 bases and encodes for minor capsid protein VP2 (Figure 2), which is located within the viral particle and thought to play a role in stabilising said capsid [12,20,21]. Moreover, it has also been suggested to play a role in capsid assembly and genome encapsidation (the enclosure of viral nucleic acid within a capsid) [21]. Lastly, ORF4, which is exclusive to MNV-1, overlaps with VP1 coding region of ORF2 and is translated during viral infection (Figure 1) [14]. Although reverse genetics experiments have shown that ORF4 is not essential for viral replication, but its encoded protein, virulence factor 1 (VF1) (Figure 1) (Table 1), has shown to increase infection via antagonising innate immune responses in vivo [14]. In addition, the 5′ end of the NoV genome is covalently linked to viral genome-linked protein (VPg) of ORF1, while the 3′ end of the genome has a polyadenylated tail (Figure 1). Studies on NoV genome transfection have shown that the viral genome alone is capable of infecting and producing intact stable viral particles both in vivo and in vitro [21].

**Table 1 viruses-14-02811-t001:** NoV Open reading frames, genes and proteins associated with viral survival and structure.

ORF	Gene	Protein	Function	References
ORF1	NS1/2	p48 N-term Protein	Regulates RdRp activity	[12,15,22]
	NS3	Nucleotide Triphosphatase (NTPase)	Assists in RNA synthesis	[12,15,23]
	NS4	p22	Antagonistic for Golgi apparatus functions	[12,15,24]
	NS5	Viral VPg Protein	Directs host machinery to favour viral protein synthesis	[12,15,25]
ORF2 ORF3	NS6	Viral Protease	Processes viral polyprotein	[12,15,25]
	NS7	RNA-dependent RNA Polymerase	Translates viral RNA	[12,15,25,26]
	VP1	Major Viral Protein 1	Binding interface for HBGA	[12,15,26,27]
	VP2	Minor Viral Protein 2	Supports viral capsid stability	[12,15,21,26]
ORF4	VF1	Viral Factor 1 Protein	Antagonizes innate immune responses	[14,21]

### 2.2. Host Susceptibility and Receptor Engagement

NoVs’ infection is a multi-step process which involves viral particle attachment, receptor engagement, endocytosis-based entry and uncoating, and release of viral genome (Figure 3) [28]. Knowledge of how human NoV (HuNoV) infiltrates cells and promotes infection is limited to unknown, but there have been many suggested models based on the application of virus-like particles (VLPs), as they are morphologically and antigenically similar to the real virus, in addition to MNV and other calicivirus, such as feline calicivirus (FCV) applications [18,23,29,30]. Since MNV was established as a model system for NoV studies, several significant discoveries have been made with regard to its genome and replication cycle. The initial step of viral attachment to the cell surface is mediated by HBGAs (complex carbohydrates found on host cells), which are thought to serve as attachment factors for NoVs (Figure 3) [30]. The interaction between NoVs and HBGAs on the surface of gut epithelial cells, mediated by the P2 subdomain, is thought to promote the concentration of viral particles on the cell surface to increase the efficiency of viral infiltration (Figure 3).

A study on immunity performed on samples obtained from Bronson Elementary School in Norwalk, Ohio revealed that susceptibility to HuNoV was contingent upon host secretory types [31]. Host secretory types were assigned based on the host’s ability to release HBGAs into their body fluids, where non-secretory types exhibited long-term immunity against HuNoV GI.1 and GII.4 genogroups [31,32]. Non-secretory types possess a genetic mutation in the FUT2 gene encoding for the FUT2 (α 1,2 fucosyltransferase) protein, which is involved in the production of diverse HBGAs from carbohydrates [33]. Unfortunately, further studies showed that HBGAs are not sufficient in conferring susceptibility to viral infection in the development of a stable HuNoV in vitro model. The current 2 system models available for HuNoV propagation use a transformed B cell line (BJAB) and stem cell derived enteroids for the HIE system, respectively [34]. Both these systems have shown restricted viral propagation, supporting only 4-passages, which in turn limits its use for detailed examination of the NoV life cycle. Recent advancement in the permissive HuNoV model cultivated in stem-cell derived non-transformed HIE has shifted the focus of NoV studies. Initially, the recommended protocol described the dispersion of 3D-HIE in a 2D-monolayer differentiated prior to infection, but was disadvantaged due to being a labour-intensive and time-consuming process [35]. In 2022, a more efficient and promising protocol was published demonstrating that differentiated 3D-HIE was prone to infection by HuNoV with increased yield, reduced experimental time, and reproducibility [35]. This model shows promise and will assist in studying the infection cycle and pathogenesis of HuNoV in more detail.

Information regarding NoV receptor engagement was unknown until 2016, when an immunoglobulin (Ig) domain-containing membrane protein, CD300lf, was discovered via genome-wide CRISPR screening. The genetic disruption depicted its essence in and sufficiency at interacting with MNV-1, inducing infection (Figure 3) [36,37,38]. Belonging to the CD300 protein family, it engages with phospholipids and operates as a cell death receptor [36]. While the receptor engaging with HuNoV is still unknown, but comparative structural studies of MNV-1, HuNoV, and FCV receptor engagement interfaces would assist in the creation of a model for the HuNoV target receptor [30]. This would also contribute towards the improvement of a HuNoV permissive model.

### 2.3. NoV Life Cycle

Due to the absence of a HuNoV permissive model, elucidating its life cycle has been a struggle. The model presented for HuNoV life cycle is based on a collection of studies performed on FCV of the Caliciviridae family, MNV gene expression and replication, and HuNoV replication systems [29,39,40,41]. Receptor engagement is thought to initiate endocytosis-based entry of NoV into its target cell (Figure 3), where viral RNA is released through endosomal uncoating [42,43]. The released NoV genome, a positive-sense (+) RNA strand, serves as a messenger RNA (mRNA) template for the synthesis of viral proteins. This process is mediated by the VPg, which is covalently linked to the 5′ end of genomic (g) and subgenomic (sg) RNA, where it acts as a cap substitute [44,45]. VPg employs a unique mechanism, where it recruits host cell translation initiation factors to preferentially translate viral RNA over host RNA, due to NoV RNA possessing short 5′ untranslated regions (UTRs). The translation of ORF1-4, results in the production of proteins which assemble into VLPs and assist in the upcoming replication cycle [46].

Once all viral proteins have been translated and post-translationally cleaved, a cytoplasmic membrane-bound replication complex is assembled as a platform for replication mediation, which includes viral RdRp, single-stranded (ss) and double-stranded (ds) RNA intermediates, viral enzymes, and host cell replication factors (Figure 4) [47]. This process commences by the formation of a dsRNA intermediate by replicating the positive-sense RNA resulting in a negative-sense strand. The negative-sense (antigenomic) RNA strand is vital as it serves as a template for the formation of positive-sense gRNA and sgRNA to be packaged into the VLPs [46]. Even though the initiation of negative-sense gRNA synthesis is not fully understood, but it is attributed to a de novo initiation mechanism (Figure 4) [48]. The de novo initiation mechanism triggers the synthesis of antigenomic RNA via a “specie-specific concentration-dependent interaction” between viral RdRp and the S domain of VP1 [48,49].

The formation of short antigenomic templates for sgRNA replication similarly commences via de novo initiation, but is thought to form in one of two ways (Figure 5) [47]. The first model suggests that short antigenomic sgRNA strands form as a result of premature termination during synthesis due to a termination signal present on the positive-sense strand [29]. The second model attributes their formation due to the existence of a highly conserved RNA stem-loop structure upstream of ORF2 on the antigenomic strand, which is thought to act as a promoter for the synthesis of positive-sense sgRNA (Figure 5) [29,50]. Bioinformatic analysis of this conserved structure confirmed that its destruction resulted in loss of replication ability, hence confirming its importance in the NoV life cycle [50]. Using MNV-1 as a model, disruption of the RNA stem-loop structure produced a 15- to 20-fold reduction in viral infectivity in comparison to control groups.

Following de novo initiation for the synthesis of antigenomic RNA, the synthesis of positive-sense RNA is thought to occur via VPg-primed initiation (Figure 4) [48]. VPg acts as a protein primer in initiating the synthesis of positive-sense gRNA and sgRNA at the 3′ end of the antigenomic RNA template, when RdRp mediates the formation of a VPg phosphodiester linkage to the initiating nucleotide [29,51]. This linkage process, VPg nucleotidylation, occurs between the conserved tyrosine residue in VPg (Y26 in MNV/Y27 in HuNoV) and the guanine amino acid at the 5′ end of the positive sense strand, and is considered essential for NoV infectivity [52,53]. This model of VPg-primed initiation is widely accepted as studies on RNA extracted from MNV-infected cells were found to be VPG-linked [51].

After the antigenomic RNA templates have undergone VPg-primed initiation and resulted in the formation of positive-sense gRNA and sgRNA, viral RNA is packaged into viral particles (Figure 4) [46]. It was mentioned earlier that basic structural studies on VP2 suggested that it might play a role in the encapsidation of RNA into viral particles, but currently this hypothesis stands untested and unconfirmed [54]. The lack of knowledge surrounding this hypothesis could be attributed to our lack of knowledge of whether VP2 possess a “structurally ordered core” which could be structurally analysed to begin with [21]. Not much is known about how NoV particles are released, but models have suggested that apoptosis-induced membrane collapse of infected cells results in its release [55,56,57].

### 2.4. Modes of Transmission

HuNoV particles spread easily due to an infected individual’s ability to shed billions of viral particles via faeces or oral mucous, which can contaminate food, water and surfaces (Figure 6) [58]. It is highly contagious due to its low infectious dose being as little as 18 viral particles, in contrast to Influenza A virus which has an infectious dose that ranges between 1.95 × 10^3^–3.0 × 10^3^ viral particles [59,60]. NoV is considered to be a seasonal disease, which spreads more easily during the winter [61]. Its tendency to spread in the winter is thought to be attributed to the rain assisting in contaminating water supplies, as well as its ability to thrive in cooler temperatures [3]. It has been suggested that NoV is able to infect oysters grown in contaminated water due to them harbouring HBGA-like structures in their tissues [62]. Ingesting contaminated oysters is thought to be one of the common modes of transmission for NoV and has been recently reported as a source of outbreak in the US [63]. In addition, it is also responsible for 16% of NoV infections caused via contaminated food in the UK yearly [64]. Direct contact with infected persons or contaminated objects almost always results in transmission of HuNoV (Figure 6), so it is important to look out for symptoms and know how to protect oneself against it [59].

### 2.5. Pathophysiology and Immune Response

In the past, incubation period estimates were not accurate due to the misclassification of secondary transmission as primary ones, but it is currently estimated to be between 10 to 51 h [65,66]. When investigating the clinical features of HuNoV, individuals could be split into asymptomatic and symptomatic types [67]. Asymptomatic NoV cases are frequent, where faecal excretion is a common occurrence, especially in children. On the other hand, symptomatic cases, which impact all age groups, involve a range of symptoms with the most common being diarrhoea, vomiting, nausea, and stomach pains, and additional symptoms being fever, headache, and body aches [58].

The two age groups facing the largest risk are younger children below the age of 5 and elderly above the age of 85, with the former being at a higher risk [68,69,70]. It is thought that these two groups are at higher risk due to their immunosenescence and acute dehydration of the elderly and young, respectively [2,71,72]. One of the other areas where knowledge regarding NoV is lacking concerns the hosts’ acute immune response. There have been advancements in the past decade on that front with regard to GI.1 and GII.4 elicited immune response suggestive of a T helper type 1 (Th1)-skewed T cell response with some Th2 activation in human subjects (Figure 7) [73,74]. While Th1 activation is proinflammatory and excessive activation can result in tissue damage of gut epithelia during infection, Th2 activation downregulates Th1 cytokines through elevated expression of IL-10 and eliciting an adaptive immune response [75]. The imbalance of Th1 and Th2 T cell activation in the acute immune response against GI.1 and GII.4 results in a proinflammatory environment, which assists in prolonging infection [73]. Evidence shows that while an innate immune response controls the spread of NoV, but an efficient adaptive immune response is required for clearance. Exploring the potential of balancing the Th1 and Th2 responses serves as a prospective therapy to overcome this obstacle, and serves to improve the available knowledge of NoV immunology (Figure 7).

## 3. Classification of NoV

The emerging need to classify different strains of NoV occurred in the mid-1990s, when RdRp partial nucleotide sequencing resulted in its division into genogroups and genotypes [76]. A few years later, classification of NoVs became dependent on complete viral protein 1 (VP1) amino acid sequencing to assign them into the different genogroups and genotypes, with a 20% threshold for sequence differences, reviewed to be 15% in a later study [77,78]. The lack of a standardised classification system for NoVs resulted in several misclassifications and inconsistencies in data over the years, which impacted the organisation of the growing knowledge of NoVs.

In 2013, the Norovirus Classification Working Group (NCWG) proposed a universal classification system of nomenclature and dual typing for NoV genotyping using phylogenetic clustering of complete VP1 amino acid sequencing and partial sequencing of RdRp [13]. The data obtained from VP1 sequencing was used for assignment to genogroups and genotypes, while RdRp partial sequencing data was used for assignment to polymerase groups (P-groups)and P-types [13,18,78]. This classification system resulted in the assignment of six genogroups (GI-GVI) and over 40 genotypes in 2013 [13]. Further advancements in NoV classification led to the emergence of a seventh genogroup (GVII) in 2015, which later expanded to 10 genogroups (GI-GX) and 49 genotypes (9 GI, 27 GII, 3 GIII, 2 GIV, 2 GV, 2 GVI and 1 genotype each for GVII, GVIII, GIX [formerly GII.15] and GX) in 2019 [79,80]. In addition to genogroups and genotypes, partial RdRp sequencing data resulted in 60 P-types, 2 tentative P-groups, and 14 tentative P-types (Figure 8).

Until recently, the MNV-1 of genogroup GV (Table 2) was the only model system available to study NoVs, before the establishment of a permissive HuNoV model in stem cell-derived HIE. Using macrophages and dendritic cell lines, significant advances in understanding NoV infection have been completed with the MNV model, including determining crystal structure of NoV, role of HBGA in infection, and the first genome sequence [39,81]. NoV strains interact with HBGAs in a strain-dependent manner, where genogroups known to infect humans include GI, GII, GVIII, and GIX (formerly GII.15) (Table 2) [13,79,80]. In recent years, the integration of norovirus diagnostic testing by real-time reverse transcription PCR (RT-PCR) in public health and clinical routines has led us to confirm the global clinical burden of norovirus genogroups GI (Norwalk virus genogroup) and GII, making them the most studied genogroups [79,82]. GII.4 is especially significant as it has caused a multitude of outbreaks globally over the years and is considered to be predominant in human infections [2]. Recently, epidemiological surveys concluded that GII.17 is emerging worldwide, with three variants identified [83].

### 3.1. GII.4 and NoV Pandemics

The first report of a GII.4 gastroenteritis outbreak was in 1987 [47]. This strain is thought to have been circulating in the human population since 1974, and is associated with more severe implications, such as mortality in contrast to non-GII.4 outbreaks [20]. GII.4 is characterised by more rapid mutation and evolution rates, which is what is hypothesised to assist in the rapid and consistent emergence of new variants every 2–4 years to take over (Figure 9) [84,85,86,87]. It has been suggested that immunocompromised patients serve as a reservoir for new NoV variant emergence, where NoV in faecal samples from patients exhibited high positive selective and mutation rates, especially within the P2 subdomain [88]. In addition, faecal excretion was shown to be far more persistent in such hosts in comparison to healthy individuals [67]. In a study observing NoV infections in hematopoietic stem cell transplant (HSCT) recipients, some patients experienced gastrointestinal graft versus host disease (GVHD), where the patients’ bodies are recognised as “foreign” [89]. In a parallel study, three patients who had acquired NoV after undergoing allogenic HSCT exhibited “villous blunting and an increased number of apoptotic CD8+ lymphocytes” as a result [90]. Their infection with NoV resulted in multiple complications resulting in their death, one with aspiration and two with sepsis [67,90].

One of the first pandemics reported to be caused by a GII.4 variant, 95/96-US variant, was in the US in 1999 [91]. The 95/96-US variant was responsible for 367 and 314 outbreaks in 1995 and 1996, respectively, reported by the Communicable Disease Surveillance Centre (CDSC) at the time. The strain dominating the US throughout the 1995–1996 season was found to have spread to 7 countries on 5 different continents, but knowledge of the source of transmission remained unknown. On the other hand, the emergence of the 95/96-US strain allowed for the implementation of an international surveillance system for tracking NoV strains.

Since the first reported GII.4 pandemic, five more pandemics have occurred (Figure 9) (Farmington Hills (2002), Hunter (2004), Den Haag (2006), New Orleans (2009), and Sydney (2012)), where the Sydney (2012) strain has dominated since [92]. There is no clear understanding of why the Sydney (2012) strain has dominated since its emergence, but it could be a result of its ability to effectively evade immune attacks. It has been reported that the emergence of a new GII.4 variant is associated with a sudden increase in gastroenteritis outbreak cases, and this has been observed in several regions globally in Europe and Asia [93,94].

### 3.2. Epidemiology

It is estimated that NoV outbreaks result in around 400,000 infectious cases, with 56 mortality incidents in the UK annually [95]. It is important to stay alert of the symptoms and report contracting NoV, as underreporting is common and impacts epidemiological estimates of the different routes the outbreak originated from, which in turn impacts control measures.

Ever since the COVID-19 pandemic commenced, there has been decreased surveillance and testing for other diseases due to the exhaustion of healthcare systems globally and their focus on SARS-CoV-2 [8,96]. This inevitably resulted in the underreporting of endemic-causing diseases, such as NoV [97]. According to Public Health England (PHE), the levels of NoV outbreaks for the 2021/2022 season were observed to be much higher than it had been for the past five seasons [9]. This is likely to be attributed to the “immunity debt” imposed as a result of the lockdown restrictions resulting in low exposure to NoV for the past three years [8]. A similar pattern of increased outbreaks has been seen recently in the US, with 103 cases reported across 13 states, all of which are linked to oysters [58]. The emergence of these outbreaks post-pandemic calls for immediate implementation of control measures and raising public awareness towards it.

Out of every five acute gastroenteritis cases, one of them is caused by NoV [98]. There are 685 million cases of NoV globally, and around 200 million of them are seen in children under the age of 5. Children form the group with both the highest incidence of NoV infection and risk of mortality with 50,000 deaths reported yearly [99]. This pattern is observed more so in developing countries, than in developed. It is more difficult to estimate the clinical burden imposed on adolescent groups and adults, but it is likely to present itself in the form of morbidity than mortality [100]. There is very little data available with regard to the aforementioned age groups, as well as developing countries and this leaves gaps in the available epidemiological knowledge of NoV.

About 40% of all NoV outbreaks occur in acute care and long-term care facilities [5]. Cases of NoV are more frequently detected in developed countries (20%), in contrast to low-mortality (19%) and high-morality (14%) developing countries [100]. One of the main differences observed in transmission patterns internationally relates to the frequency of outbreaks in acute-care and long-term care facilities. While these environments are attractive for NoV outbreaks and detection, but the same could not be said for low-and-middle- income countries (LMICs) as a result of “absence of diagnostics and lack of surveillance”, as well as a small population of elderly occupying long-term care facilities. This is mainly considered to be a result of both the difficulty of routinely detecting HuNoV (non-cultivatable) and the prominence of other pathogens in developing countries.

Healthcare facilities create attractive environments for such outbreaks due to the ease of spread among the present vulnerable populations, such as the institutionalised elderly and immunocompromised persons [101]. Ease of transmission results in ward closure, which financially burdens healthcare facilities [5,102,103,104].

One of the main risk factors for NoV nosocomial outbreaks is attributed to spatial proximity, as hospital environments have high levels of contact, which contributes to the ease of transmission [105]. It is common for patients’ beds to be grouped in spatial proximity in the same bay, and during NoV outbreaks this organisation contributes towards increasing the ease of transmission among those patients, in addition to compromising patient safety. A study on an outbreak which occurred in a US tertiary care hospital found that 104 healthcare workers from two separate units were infected, which reinforces the role of spatial proximity and high-level of contact in spreading NoV [106]. Another important factor to understanding the transmission of NoV in healthcare settings is identifying the drivers of transmission. A modelling study performed on Dutch hospitals found that the main drivers of transmission were symptomatic patients and not asymptomatic workers as would have been expected [107]. When comparing symptomatic shedders to asymptomatic shedders, the latter were found to be non-infectious regardless of shedding unlike symptomatic shedders. These findings suggested that in asymptomatic cases, transmission is more dependent on proper personal hygiene rather than anything. The lack of a vaccine against NoV augments both the economic and clinical burden. Evidence shows that the development of a vaccine with 50% efficiency is estimated to prevent 2.2 million cases annually in the USA, highlighting the impending need for a vaccine [108].

## 4. Current Hand Hygiene Methodologies for Combating NoV Transmission

NoV is a global health issue, which has the ability to add untold pressures to healthcare services, as its lack of specialised control and prevention measures makes NoV troublesome and difficult to manage. Advise recommended by governmental bodies, such as the Healthcare Infection Control Practices Advisory Committee (HICPAC), for future NoV outbreaks provide a certain level of protection against NoV, but in practise are basic control measures advocated for multiple pathogens, mainly targeting decreasing the transmission rate in order to provide protection. The measures could be divided into multiple categories but the most important ones would be contact precaution, clean-up, and hand hygiene, alongside ward closure being the most controversial as it imposes a high financial burden [109,110].

According to HICPAC, contact precaution advises for patients to be placed in single occupancy rooms, but when it cannot be accommodated then asymptomatic and symptomatic patients should be separated [110]. In addition, contact precaution to be enforced for a minimum 48 h post symptom resolution. During and after an outbreak, clean-up advice recommends the use of bleach-based products and performance of routine cleaning of frequently touched areas starting with areas with a lower likelihood of contamination to areas with high likelihood of contamination [111].

One of the main protective measure indicated worldwide for the prevention and control of NoV is good hand hygiene. Interestingly, there is very little data on the effects of hand wash products commonly used commercially or within a healthcare setting on NoV, however some studies have been conducted. Liu et al. (2009) conducted a study on the effectiveness of liquid hand soaps and sanitisers on NoV on a small cohort of 10 volunteers with contaminated hands. In brief, individuals were inoculated with 5.6 × 10^6^ NoV genomic copies per finger pad. Two thumb pads were used per experiments to generate a baseline at t = 0 and one at t = 20 s exposure after initial treatment. Inoculation was allowed to dry before the addition of 1.0 mL of hand sanitizer or liquid soap at differing concentrations. After which, skin cells and dried inoculation were scraped, eluted with water and then inverted with a Hanks balanced salt solution (HBSS) to remove any excess NoV. RT-PCR was used to determine viral copy numbers as a mode of verifying viral survival. Results suggest that anti-microbial liquid hand wash demonstrated the greatest efficacy in NoV killing with a 0.67 to 1.20 log_10_ reduction and the alcohol base hand sanitisers (62% ethyl alcohol content) by 0.14 to 0.34 log_10_ reduction, rendering the hand sanitiser relatively ineffective. Interestingly in an additional suspension study, whereby sodium hypochlorite (a chemical compound actively found in bleach) at a concentration of ≥160 ppm was capable of effectively neutralizing all NoV particles [112].

Unfortunately, the corrosive nature of sodium hypochlorite makes it an unfit candidate for hand wash formulas, likely resulting in damage to the skin barrier. Many of the antimicrobial properties found within hand washes are attributed to surfactants. Surfactants are amphiphilic molecules that contain both hydrophobic and hydrophilic parts and are usually the component of soaps and disinfectants which provide their anti-microbial properties and help destabilise viral structures. Predmore and Li (2011) results demonstrated that surfactants such as sodium dodecyl sulphate (SDS), Nonidet P-40 (NP-40), Triton X-100, and polysorbates were all capable of reducing viral titres, however, in accordance with both the British and European standards these surfactants alone or in combination were not deemed anti-viral [113]. This is because a 4-log reduction from the original viral titre needs to be achieved in order to achieve anti-viral status. Data such as these supports previous works such as Liu et al. (2009) [112]. However, this study was conducted in a food industry setting and not in a clinical setting and therefore results should be interpreted accordingly.

Even though previous works have shown most readily available wash products and alcohol based hand sanitiser to be relatively ineffective, research still needs to continue into finding innovative formulations or regimes in order to neutralise NoV and reduce its transmission rates. With limited success being found within hand sanitising products, alternative approaches such as anti-viral therapies and vaccines are currently undergoing clinical trials with relative success. This will be discussed in the following sections.

## 5. Future of NoV Treatment

Ever since the application of VLPs has been introduced in studying NoVs, they have formed an attractive vehicle for vaccine development. There are currently many vaccine candidates in development and clinical trial phases, and the majority of them administer VLPs. One of the main challenges faced in the development of a NoV vaccine is their heterotypic nature and ability to mutate rapidly (Table 3) [18]. This highlights the need for a NoV vaccine to be bivalent against the two main infectious genotypes GI.1 and GII.4. Studies have shown that immunisation against one genotype also provided immunity against other genotypes, which could allow us to conclude that a bivalent vaccine protecting against more than one NoV variant is possible [18,114,115].

The emergence of a new strain of GII.4 every 2–4 years also presents an issue when developing a vaccine. This is a similar issue as the one faced by Influenza virus, and would require for the vaccine to evolve as the new variants emerge [116]. An ideal NoV vaccine would protect against multiple variants asides from what it was designed for, and the possibility of that is likely as suggested in the previous paragraph. One vaccine developer suggested a strategy which would involve the development of a vaccine against the major capsid protein of a combination of three GII.4 variants [117]. In a randomized, double-blind, placebo-controlled trial (63 vaccine and 64 placebo), immunisation with such vaccine resulted in acquired immunity against GII.4 present at the time (September 2010 to April 2011) and variants that appeared years later as well [117].

While there is still very little known about the ideal method of developing a NoV vaccine, but there are several candidates on the market (Table 3) [118]. The most studied vaccine candidate is the HIL-214 (TAK-214 previously) candidate, which is a VLP-based bivalent vaccine that employed the design strategy suggested by Atmar and colleagues [117,118,119]. HIL-214 is made of both GI.1 and GII.4 VLPs, which are the two dominant genotypes in causing human gastroenteritis. It has progressed through several stages of clinical trials and has undergone phase 2b trials. The conclusions of phase 2b trials showed that it was capable of providing cross protection, hence making it a promising candidate [120]. While the other candidates employ similar design concepts and show good potential, but there are still several stages they have to undergo before a clear decision about them could be made. The developing process of the ideal NoV vaccine is still underway and while many problems are faced, but the knowledge gap is diminishing every day and increasing the possibility of finding a way to combat NoV and ease the clinical and financial burden it imposes globally.

Another wave of NoV treatments employs small molecule inhibitors targeting viral and host co-factors, as well as NoV structural and non-structural proteins. The application of the MNV cell model assisted in the collection of X-ray crystallographic data of viral RdRp AND 3CL-pro (protease), as well NMR spectroscopy and computational docking experiments, all of which have assisted in the identification of anti-NoV molecules [121]. Some antiviral drugs target the interaction between VP1 and HBGA, such as L-fructose inhibitors, while others focus on targeting RdRp activity or NoV 3CL-protease activity [122]. Unfortunately, none of the studied anti-NoV drug candidates reached clinical trials due to exhibiting low bioavailability in vivo. This is owed to the rapid metabolism of these drug molecules by multiple proteases, hence hampering their impact. Until the efficiacy and bioavailability of small molecule inhibitors could be improved, antivirals prospect does not out-compete with vaccine candidates. On the other hand, nitazonxanide, an FDA approved drug originally developed as an anti-protozoal agent, has shown to aid in reducing the duration of resolution of symptoms and is being applied to several NoV patients.

**Table 3 viruses-14-02811-t003:** NoV Vaccine Candidates.

Company	Vaccine Candidate	Administration	Antigen Form: Genotype	Clinical Trial Phase	Reference
Takeda	HIL-214 (previously TAK-214)	Intranasal, Intramuscular	NoV VLP: GI.1/GII.4	Phase 2b	[123]
Vaxart	VXA-NVV-104	Oral	Adenovirus-expressing NoV VP1NoV VLP: GI.1/GII.4	Phase 1	[124]
NVSI	Hansenulapolymorpha	Intramuscular	NoV VLP: GI.1/GII.4	Phase 1	[125]
IPS/Zhifei	Longkoma	Intramuscular	NoV VLP: GI.1/GII.3/GII.4/GII.17	Phase 2a	[126]

## 6. Conclusions

Without a doubt, NoVs continue to cause epidemic and endemic acute gastroenteritis with significant morbidity and mortality worldwide, especially with current pressures on healthcare services, causing a reduction in the NoV surveillance and increase in transmission. NoV genotype GII.4 continues to dominate worldwide, with PHE reporting a significant increase in NoV cases from 2021/2022, compared to the previous five season, with current restrictions resulting in an ‘immunity debt’ throughout society. Just under half of all NoV reported cases occur in acute care and long-term care facilities, making healthcare services an attractive environment for NoV, especially as many people within such facilities are in close-contact and immunocompromised. Unfortunately, many of the control measures still in place today, as advocated by health organisations and governments still rely on basic methods such as good hand hygiene practises and regular cleaning/disinfection of contaminated areas. However, recent data has shown a lack of protection from readily available hand hygiene products and hand sanitizers, with only harsh chemicals such as sodium hypochlorite (used within bleach) being an effective disinfectant. This would be acceptable for sanitizing surfaces, but not for hand hygiene purposes, due to its corrosive nature. These conclusions allow us to pose the question, “how can we better protect the public from hand-to-hand transmission of viruses such as NoV?”. Currently, the need for new hand wash products or alternative medicines (vaccines etc.) is of utmost importance. The current success in the bivalent mRNA COVID-19 vaccination programme, has proven to be highly protective, as well as its further applications in developing vaccine candidates against influenza and malaria. These support the development of similar vaccines for NoV. Following the footsteps of COVID-19 vaccine development, NoV VP1 or even its receptor binding P-domain which is similar to the spike protein of SARS-CoV-2 could be an excellent candidate for an mRNA vaccine against NoV infection. Besides vaccines, small molecule inhibitors also provide an attractive alternative method of controlling the spread of NoV and providing protection from future epidemics. Continuing research provides hope for future prevention of and protection against NoV outbreaks worldwide.

## Figures and Tables

**Figure 1 viruses-14-02811-f001:**
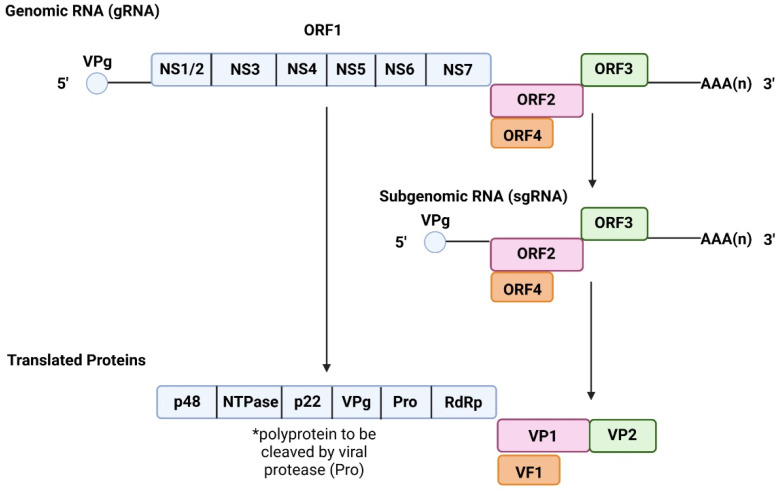
NoV Genome Organisation and Translated Proteins. NoV is divided into ORF1-ORF4, where ORF4 is only present in MNV-1. ORF1 encodes for a structural polyprotein, which when cleaved by viral protease gives rise to 6 non-structural proteins. ORF2 encodes for major capsid protein VP1 and ORF2 encodes for minor capsid protein VP2, which both form the viral capsid. VF1 encoded for by ORF4 in MNV-1 has been shown to assist in enhancing viral infectivity. Made using Biorender.com, accessed on 18 October 2022.

**Figure 2 viruses-14-02811-f002:**
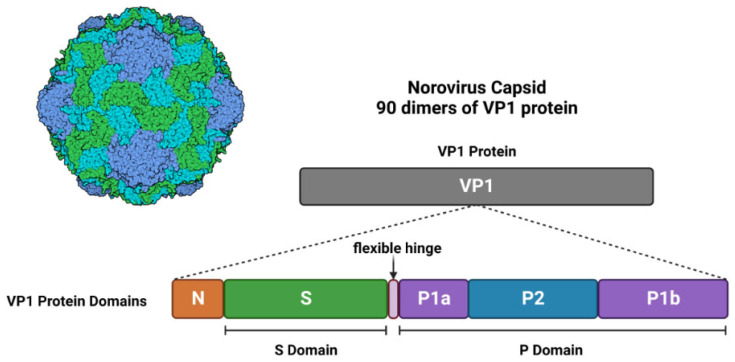
NoV Viral Particle Structure. NoV capsid is formed of 180 VP1 proteins assembled into 90 dimers. The P2 domain of VP1 is the most surface exposed part of the capsid and is highly variable. This interface is thought to be the one which interacts with HBGAs. Made using Biorender.com, accessed on 18 October 2022.

**Figure 3 viruses-14-02811-f003:**
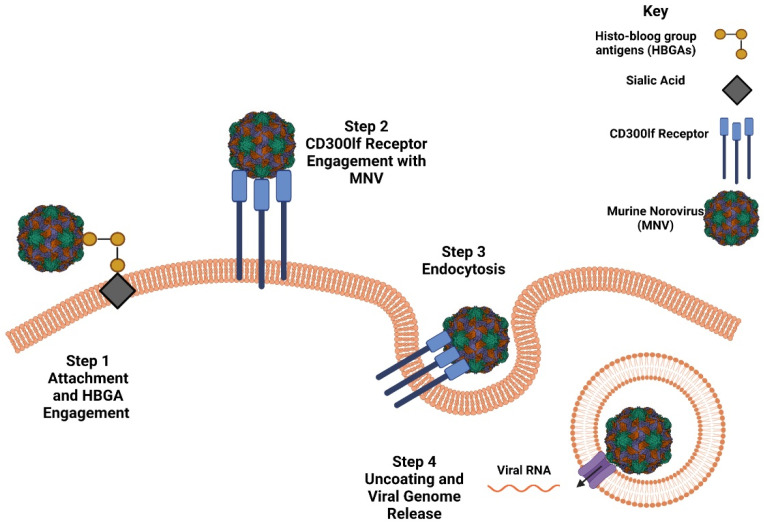
HBGA and Receptor Engagement with MNV-1. HBGA act as attachment factors for MNV-1, where it improves infectivity by concentrating MNV-1 on the membrane. Recently, CD300lf, a protein belonging to the CD300 family, was identified as receptor which interacts on the surface of murine macrophages with MNV-1. It belongs to a family of cell death receptors and it has proven to be essential for viral ability to infect and replicate. Interaction with CD300lf results in the initiation of viral uptake via endocytosis, which is followed by uncoating and viral genome release. Made using Biorender.com, accessed on 18 October 2022.

**Figure 4 viruses-14-02811-f004:**
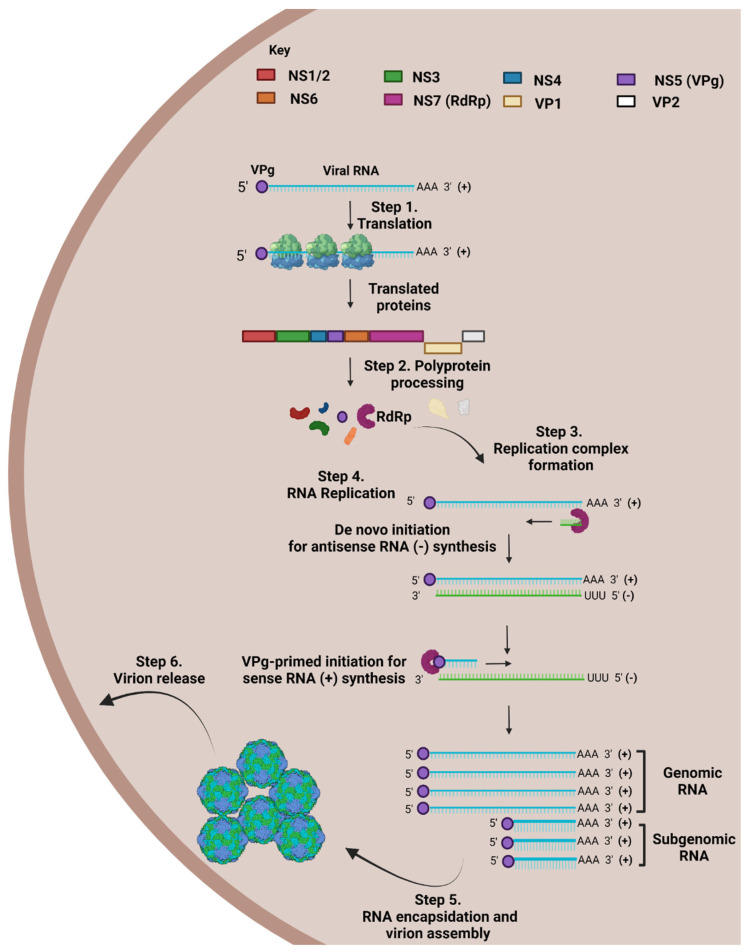
NoV RNA replication model. NoV genome is a positive sense RNA strand, which is translated to give rise to NoV proteins. The replication of the positive RNA strand occurs through two initiation steps. De novo initiation occurs at the 3′ of viral genome resulting in the formation of an antisense RNA strand (-) using RNA-dependent RNA-polymerase (RdRp) NS7. This antisense strand is used as a template for the formation of a sense through VPg-primed initiation using RdRp again. Resulting genomic and subgenomic viral RNA undergoes encapsidation into virions, which are released from the cell. Made using Biorender.com, accessed on 18 October 2022.

**Figure 5 viruses-14-02811-f005:**
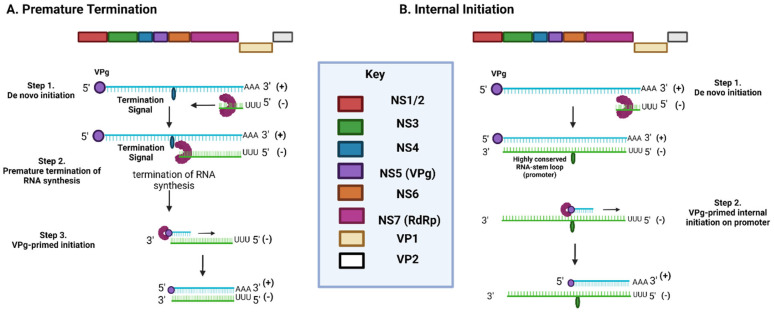
Two models for subgenomic RNA formation. The premature termination model suggests that a termination signal within genomic RNA results in early termination during de novo initiation replication. On the other hand, the internal initiation suggests that a highly conserved RNA-stem loop upstream of ORF2 is responsible for early termination of de novo initiation replication. The short antisense RNA strands produced then undergo VPg-primed initiation to form positive sense subgenomic RNA. Image made in Biorender.com, accessed on 18 October 2022.

**Figure 6 viruses-14-02811-f006:**
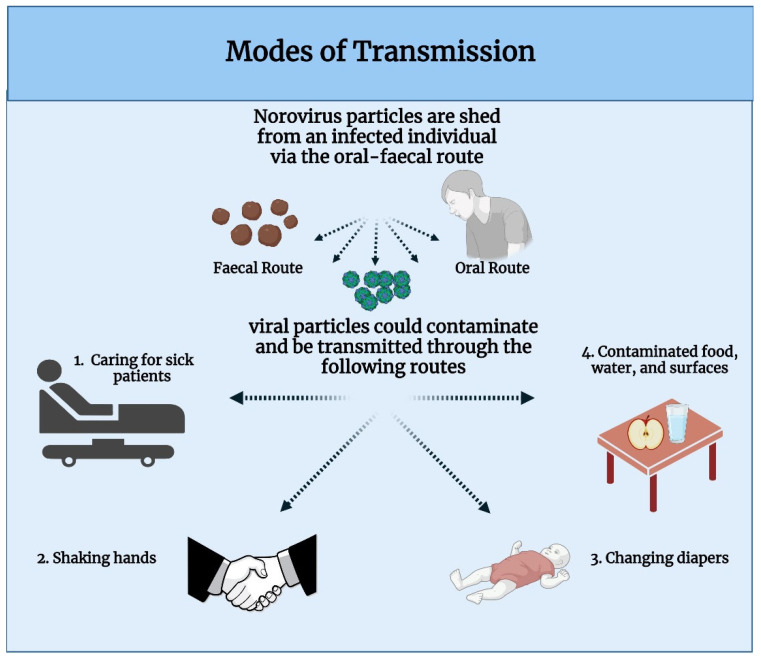
Transmission Modes of NoV reported by the CDC. Image made in Biorender.com, accessed on 18 October 2022.

**Figure 7 viruses-14-02811-f007:**
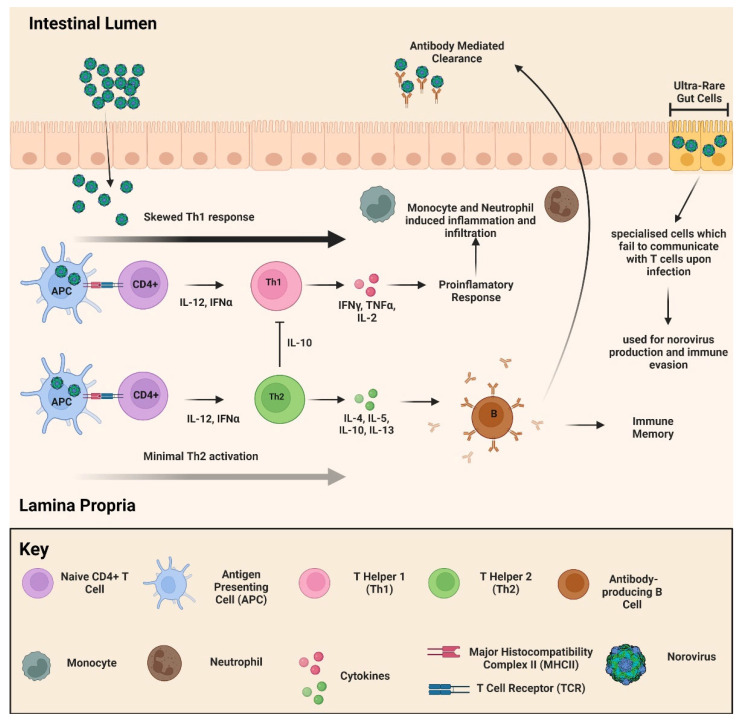
A schematic overview of immune activation to NoV infection. Diagram indicated both Th1 bias immune activation to NoV infection (dominant immune activation), alongside the Th2 activation. Figure made in Biorender.com, accessed on 18 October 2022.

**Figure 8 viruses-14-02811-f008:**
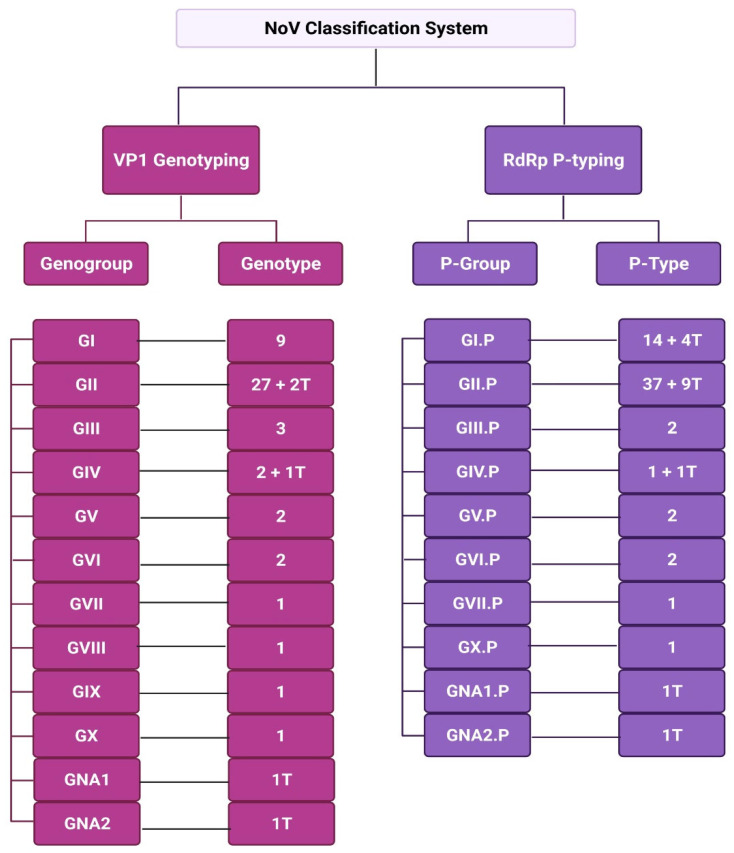
NoV Classification. NoV variants are classified on the basis of whole amino acid sequencing of VP1 capsid protein for assignment into genogroups and genotypes, while partial RdRp sequencing is used for assignment into P-groups and P-types. Currently there are 10 genogroups. Image generated using Biorender.com, accessed on 18 October 2022.

**Figure 9 viruses-14-02811-f009:**
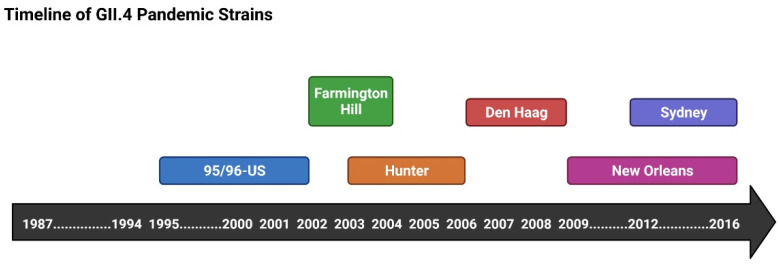
Timeline of GII.4 Pandemic Strains. The first GII.4 pandemic was reported in 1987 and was caused by the 95/96-US variant. Ever since, every 2–4 years a new variant emerged to take over. Since then, five more pandemic strains have emerged, and the Sydney (2012) variant has dominated ever since. Made using Biorender.com, accessed on 18 October 2022.

**Table 2 viruses-14-02811-t002:** NoV Genogroups and Hosts [13,79,80].

Genogroup	Host
GI	Human
GII	Human and Porcine
GIII	Bovine
GIV	Human and Feline/Canine
GV	Murine
GVI	Canine
GVII	Canine
GVIII	Human
GIX	Human
GX	Bat

## Data Availability

Not applicable.

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
