# Peer review of "Norovirus: An Overview of Virology and Preventative Measures"

_viruses, 2022, doi:10.3390/v14122811_

Round 1

Author Response

I would like to say thank you for taking the time to review the review, and we value the comments and suggestion given. 

Reviewer 2 Report

This paper systematically reviewed Norovirus (NoV) including its evolution, virology, current prevention practices (hand hygiene) and the future treatment. The topic fits the scope of Viruses, and will benefit the understanding, preventing as well as the further treatment of NoV. In general, the manuscript is well-organized and the references are supportive to the conclusions, while key issues are required to be addressed before its publication on Viruses.

Major points:

1. In the introduction part, the seasonal feature of NoV outbreak is required to be discussed.

1. In section 4, the other prevention strategies, such as clean-up and isolation, are required to be discussed.

2. In section 5, the development of small-molecule drugs targeting various key proteins of NoV is required to be discussed.

Minor points:

1. The section 2.0 should be 2, or some general summary words should be there.

2. In Table 1, the references for the genogroups are same and may be noted in the Table title. Optionally, the host information may be combined with the classification in Fig1.

3. In Table 2, the key functions of each protein are suggested to be summarized and put in this table.

4. In Fig 6 and 7, the font size of labels is too small and needs to be adjusted.

5. In section 3.4, line 5, the format of viral particles number is incorrect.

6. On page 15, section 4, the 2nd line from the bottom, the format of gnomic copies number is incorrect.

Author Response

(The authors gave the same response as above.)

Round 2

Reviewer 1 Report

The revised article “Noroviruses: a history of outbreaks and preventive measure” is more understandable and clearer than the previous version.

I suggest only minor corrections that should be addressed before publication.

Minor Revision

In general, all references in the text should be placed in square brackets as required by the journal. Check all text.

Abstract

Line 19, line21 and line 27, remove line breaks

Keywords: keywords are missing. please add them

Introduction

Line 90. Insert a space between polymerase and (RdRp)

Line 92 Table 2 is wrong, I think it is table 1

Chapter 2.3  

Line 202: Insert a space between 2.3 and NoV

Chapter 2. 4. line 294: Figure 8 is wrong, I think it is Figure 6

Pag 17 line 525: you wrote “eluted with water and then inverted in l Hanks balanced salt solution (HBSS)”. Something is missing before or after l, it is not clear, please clarify.

Line 544: statues??

I think it’s wrong. Please check it.

Chapter 5.

line 560. Move the reference to Table 3 after the VLPs on line 559.

Remove also table 3 in line 566

Table 3: Insert a space between table and 3

Conclusions

There are some typos:

Line 616: survaillence

Line 622: Unfortueatley

Line 626: santisiers

Line 632: prodcuts

Line 639: moluecle

Please correct them.

And line 631 remove line breaks

Author Contributions, the authors names must be replaced with initials. Please see author guidelines.

References

The formatting of the bibliography is not correct. For example: the initials of the names must have a dot, the year of publication must be in bold. the name of the journal must be abbreviated and in italics. Please see author guidelines.

Author Response

In general, all references in the text should be placed in square brackets as required by the journal. Check all text.

  • All references have been placed in square brackets  

Abstract

Line 19, line21 and line 27, remove line breaks

  • All lines breakers have been removed

Keywords: keywords are missing. please add them

Introduction

  • Thank you for highlighting this oversight. The key words have now been included into the review

Line 90. Insert a space between polymerase and (RdRp)

  • A space has been inserted

Line 92 Table 2 is wrong, I think it is table 1

  • The correct table has been added to the corresponding text

Chapter 2.3 

Line 202: Insert a space between 2.3 and NoV

  • Space has been inserted

Chapter 2. 4. line 294: Figure 8 is wrong, I think it is Figure 6

  • The correct figure has now been included into the corresponding text

Pag 17 line 525: you wrote “eluted with water and then inverted in l Hanks balanced salt solution (HBSS)”. Something is missing before or after l, it is not clear, please clarify.

  • The sentence has been restructured

Line 544: statues??

  • This typo has now been amended

Chapter 5.

line 560. Move the reference to Table 3 after the VLPs on line 559.

Remove also table 3 in line 566

  • The reference to the table has been moved along with the removed of table three reference on line 566

Table 3: Insert a space between table and 3

  • The correct spacing has been added to table 3

Conclusions

There are some typos: 

  • All typos have been addressed within the concluding remarks

Author Contributions, the authors names must be replaced with initials. Please see author guidelines.

  • Authors names have been replaced with initials

References

The formatting of the bibliography is not correct. For example: the initials of the names must have a dot, the year of publication must be in bold. the name of the journal must be abbreviated and in italics. Please see author guidelines.

  • Formatting has been corrected in order to fit the guidelines

We would like to thank the reviewer for taking the time to read and edit this review for a second time